# A systematic review and meta-analysis of the impact of vaccination on prevention of long COVID

Rhiannon Green [1], Zoe Marjenberg[1], Gregory Y. H. Lip[2,3], Amitava Banerjee [4,5], Juan Wisnivesky[6], Brendan C. Delaney [7], Michael J. Peluso [8], Elke Wynberg[9] & Sultan Abduljawad [10] ✉

Long COVID affects millions worldwide and its prevention is a critical public health strategy. While prior analyses show primary vaccination prevents long COVID in subsequent infections, the effect of booster vaccination on long COVID after Omicron infections is unclear. This systematic review identifies 31 observational studies, of which 11 are suitable for pairwise meta-analyses. The pooled odds ratio (OR) of long COVID in those vaccinated (any dose) versus unvaccinated is 0.77 (95% confidence interval [CI] 0.70–0.85; $p < 0.0001$; 10 studies). ORs were also lower for primary course vaccination versus unvaccinated (OR 0.81; 95% CI 0.79–0.83; $p < 0.0001$; 3 studies), booster vaccination versus unvaccinated (OR 0.74; 95% CI 0.63–0.86; $p = 0.0001$; 4 studies), and booster vaccination versus primary course vaccination (OR 77; 95% CI 0.65–0.92; $p = 0.0044$; 3 studies). These findings indicate that booster vaccination can provide additional protection against long COVID, highlighting the importance of seasonal vaccination against new SARS-CoV-2 variants. They should, however, be interpreted cautiously, given the small number of studies and the low quality of evidence.

In 2022, the World Health Organization (WHO) estimated that roughly 10–20% of individuals experience persistent symptoms following the acute phase of SARS-CoV-2 infection[1]. More recent reports, from 2024, place the proportion at 2–7%[2,3], suggesting potential era-related and vaccination effects[4]. This condition is known as long COVID and, in certain contexts, is referred to as post-acute sequelae of COVID-19 (PASC), post-COVID condition (PCC), or post-acute COVID-19 syndrome (PACS)[5,6]. The US National Academies of Sciences, Engineering, and Medicine recently published a consensus definition for long COVID as "an infection-associated chronic condition that occurs after SARS-CoV-2 infection and is present for at least 3 months as a continuous, relapsing and remitting, or progressive disease state that affects one or more organ systems"[7].

Long COVID is a multifaceted disease with differing pathologies that may result in overlapping symptoms. Symptoms can manifest in almost any aspect of physical or mental health, making management more difficult[2,8]. Living with symptoms of long COVID can negatively impact work productivity and social relationships, contributing to a general decline in quality of life[9–11]. In particular, fatigue, cognitive dysfunction, insomnia, and inability to exercise have been reported as

[1]Maverex Ltd, Newcastle Upon Tyne, UK. [2]Liverpool Centre for Cardiovascular Science at University of Liverpool, Liverpool John Moores University and Liverpool Heart and Chest Hospital, Liverpool, UK. [3]Danish Center for Health Services Research, Department of Clinical Medicine, Aalborg University, Aalborg, Denmark. [4]Institute of Health Informatics, University College London, London, UK. [5]Department of Cardiology, Barts Health NHS Trust, London, UK. [6]Division of General Internal Medicine, Icahn School of Medicine at Mount Sinai, New York, NY, USA. [7]Department of Surgery and Cancer, Imperial College London, London, UK. [8]Division of HIV, Infectious Diseases, and Global Medicine, University of California, San Francisco, San Francisco, CA, USA. [9]Mahidol-Oxford Tropical Medicine Research Unit (MORU), Mahidol University, Bangkok, Thailand. [10]BioNTech UK Ltd., London, UK. ✉e-mail: sultan.abduljawad@biontech.co.uk

some of the symptoms with greatest burden by participants in a cross-sectional survey study (n = 5163) conducted in an online COVID-19 support group[11]. A previous meta-analysis reported that across studies, an average of 52% of people with long COVID (defined as persistent effects after 3 months) had a long-lasting decrease in health-related quality of life (HRQoL), with a mean follow-up of 4.5 months[9]. This decline in HRQoL was exacerbated in participants who experienced severe COVID-19 compared with those with milder disease[9]. In relation to overall disease burden, long COVID was found to account for 80.4 and 642.8 daily adjusted life years per 1000 non-hospitalized and hospitalized persons, respectively[6]. Further, long COVID has lasting economic implications and places a significant burden on healthcare systems, largely due to hospitalization costs and healthcare visits[12]. Al-Aly et al. (2024) reported a conservative estimate based on available data that places the annual global economic cost of long COVID at $1 trillion, which equates to 1% of the global gross domestic product for 2024[13].

Currently, there are no validated, effective treatments for long COVID, and management is largely symptomatic[5,7]. This places additional value on preventative measures. Vaccination prior to SARS-CoV-2 infection has been shown to reduce the risk of developing long COVID in observational studies[14–21]. This is likely due, at least in part, to the protective effect of vaccination against severe acute COVID-19 and COVID-19–related hospitalization. Booster and seasonal vaccines have been introduced to provide greater protection against emerging SARS-CoV-2 variants containing escape mutations and reduce the risk of severe COVID-19. It follows that booster vaccination may have added benefits in preventing long COVID. Indeed, studies have shown that the effectiveness of vaccination prior to infection in protecting against long COVID is increased with additional booster doses received[22,23].

Omicron (B.1.1.529) was identified in November 2021 and designated a variant of concern by the WHO[24]. This variant has demonstrated a distinct genomic signature compared with the original Wuhan SARS-CoV-2 strain. Omicron variants, including currently circulating subvariants, display increased transmissibility and infectivity. Evidence suggests an increased ability to evade humoral immunity, suspected to be driven by its extensive mutational profile, resulted in a surge in Omicron subvariant infections throughout 2022[25]. More recently, in the summer of 2024, prevalence of the Omicron subvariant JN.1 surged, and at the time of writing, circulating JN.1-derived variants were dominant globally[26].

An up-to-date and comprehensive understanding of the impact of vaccination on the prevalence and risk of long COVID is needed to reflect evolving SARS-CoV-2 strains and changing global immunity due to evolving subvariants, reinfection, and seasonal vaccination. The majority of previous systematic literature reviews investigated the impact of primary course vaccination versus no vaccination or included non-Omicron variants in the analysis[19,20,27,28]. Therefore, there is a gap in understanding the effect of additional booster doses on the risk of developing long COVID in the context of more recent Omicron-variant data. This systematic review aimed to investigate the impact of COVID-19 vaccination, including booster versus primary course vaccination, on preventing long COVID caused by subsequent Omicron variants infection.

## Results

### Overview of included studies
In total, 9107 records were identified through searches. Following elimination of duplicates, 5360 studies underwent title and abstract screening, of which 368 full-text articles were assessed for eligibility. Following this, 31 studies were selected for the review, including two additional studies identified by hand searching (Fig. 1).

Study characteristics are summarized in Supplementary Table 1. The studies were performed in 16 different countries across Europe, North America, Asia, and Australia. Eleven studies were conducted

retrospectively[22,29–37], 17 were prospective[10,38–53] and three were cross-sectional[54–56] in design. Data sources varied across the studies and included medical centers and healthcare systems (n = 7)[30,32,33,35,41,42,55], cohort studies and registries (n = 19)[22,29,34,36,37,39,40,43–52,56,57], COVID-19 testing sites (n = 3)[10,38,53] electronic health records (n = 1)[31], and online surveys (n = 1)[54]. The definitions of long COVID adopted by the studies varied and are reported in Supplementary Table 2.

Most studies identified in the systematic literature review included adults only (27 studies)[10,22,29,30,32–35,37–43,45–53,55–57]. One study included adolescents (≥15 years old) and adults[44], and three studies included children and adolescents only[31,36,54]. Of the adult-only studies, the mean age ranged between 36.1 (standard deviation [SD] 7.6) and 61.4 (SD 16.1) years old, and the median ranged from 37 (interquartile range [IQR] 23.0, 48.0) to 66 (IQR 54, 77) years old. The percentage of female participants ranged from 12.7% to 82.0%.

Study population sizes of Omicron-infected participants ranged from 80 to 7,998,854. The Omicron infection period ranged from 13 days to 15 months. Only seven studies reported the specific Omicron subvariant present[29,41,42,49,53,54,57] and these included BA.1, BA.2, BA.4, BA.5, and XBB; in studies where subvariants were not reported, most Omicron infections occurred when BA.1/2 and BA.4/5 were the dominant subvariants. The severity of SARS-CoV-2 infection was reported by 13 studies, where hospitalization rate ranged from 0.1% to 10.3% and the rate of intensive care requirement ranged from 0.02% to 13.8%, across study cohorts. No studies stratified outcomes by acute infection severity.

Of the studies that recorded vaccine types, the following vaccines were reported: BNT162b2, mRNA-1273, Ad26.COV2.S, and ChAdOx1. The most frequently reported vaccines used were mRNA vaccines. Four studies included only participants who had received the BNT162b2 vaccine[10,36,38,53], three included a mixture of mRNA vaccines only[31,40,49] and 10 reported inclusion of a mixture of EU-authorized vaccines. Two studies included almost entirely mRNA vaccines apart from two patients who received Sinovac[54] and one patient who received Ad26.COV2.S[39], respectively. Twelve studies did not report vaccine type but were assumed to include a mixture of EU-authorized vaccines as they were performed in countries that only used EU-authorized vaccines at the time of the study. Only one study reported that participants received bivalent booster vaccination[53].

### Risk of bias
Risk of bias was found to be either low or medium (20 and 8 studies, respectively) in all studies assessed using the Newcastle-Ottawa Scale (NOS) (Supplementary Table 3). Three cross-sectional studies were analyzed for potential risk of bias using the Joanna Briggs Institute (JBI) Critical Appraisal Checklist for Analytical Cross-Sectional Studies, with two low risk[54,56] and one potentially high risk due to a lack of reporting and accounting for confounding factors[55].

### Vaccination status
This review classified vaccination status reported by the included studies (full details in Supplementary Table 4) into unvaccinated, primary course vaccination only, and booster vaccinated groups. Study participants were generally considered unvaccinated if they had not received any vaccine doses; however, one study also considered participants to be unvaccinated if they received their last monovalent dose more than 12 months before study enrolment[53]. Most studies did not report the average time between vaccination and SARS-CoV-2 infection (Supplementary Table 4).

### Vaccination impact on long COVID
Reported long COVID outcomes assessed by this review were symptom burden (mean number of symptoms) of long COVID (five studies), the prevalence or incidence of long COVID (18 studies), and the risk of long COVID (20 studies). Overall, participants who received primary

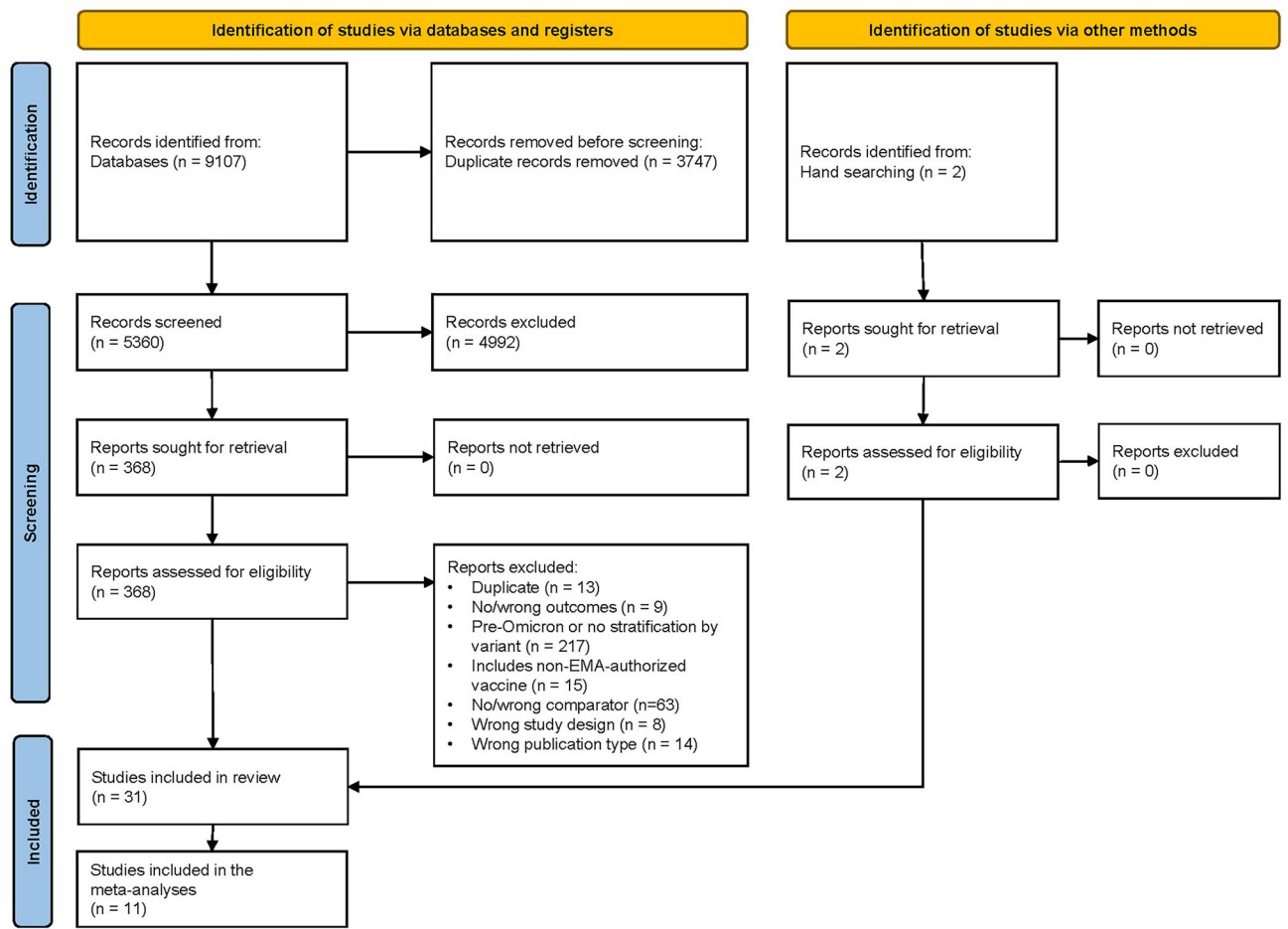

**Fig. 1 | PRISMA flow diagram.** The figure illustrates the PRISMA (Preferred Reporting Items for Systematic Reviews and Meta-analyses) flow diagram showing the number of records identified, screened, included in the review, and included in the meta-analysis. EMA European Medicines Agency.

course vaccination, booster vaccination, and additional booster vaccine doses generally had a numerically or statistically significantly lower number of long COVID symptoms, lower prevalence, and lower risk of long COVID compared with either those who received no vaccination or vaccination with a lower number of doses. The long COVID outcomes reported by these studies are presented in Supplementary Tables 5–9.

### Meta-analyses: Impact of vaccination on prevention of long COVID caused by Omicron variant infection

The feasibility assessment identified 11/31 studies[29,34,37–39,43,47–49,53] that could be included in meta-analyses of the impact of vaccination on the risk of developing long COVID after Omicron infections (Supplementary Results: 2.1 Meta-analysis feasibility assessment and Supplementary Table 10).

**Impact of vaccination prior to Omicron variant infection on long COVID compared with no vaccination** Ten studies were included in a pooled analysis of the risk of long COVID after vaccination (any number of doses) compared with no vaccination.

As three studies[10,32,35] reported separate results for both primary vaccination and booster vaccination, we included only their primary course estimates in the main analysis. We then repeated this analysis, substituting the primary course estimate in these three studies for their booster dose estimate as a sensitivity analysis to assess the impact of these alternate outcomes on the pooled odds ratio (OR). Vaccination was associated with a significantly lowered pooled OR of long COVID compared with unvaccinated cohorts (OR 0.77 [95% confidence interval (CI) 0.70–0.85; p < 0.0001]; Fig. 2). Pooled ORs were

similar in the substitution sensitivity analysis (OR 0.71 [95% CI 0.63–0.79; p < 0.0001]; Fig. 3). Between-study heterogeneity for both analyses was high ($I^2$ of 72.4% for the main analysis and 74.8% for the sensitivity analysis).

The results of the Egger's test of publication bias[58] were non-significant for both analyses, and funnel plots were generally symmetrical (Supplementary Fig. 1 and 2), indicating no evidence of publication bias.

Further sensitivity analyses indicated that the results of the meta-analyses were robust. Leave-one-out plots for both the main and sensitivity analyses did not result in loss of statistical significance of the pooled OR values (Supplementary Fig. 3 and 4). Removal of a potential overlapping population between studies, children/adolescent-only studies, studies with a high hospitalization for acute illness rate, and pre-prints for both analyses did not result in loss of statistical significance of the pooled OR values (Supplementary Tables 11 and 12).

**Impact of primary course vaccination prior to Omicron variant infection on long COVID compared with no vaccination.** Three studies[10,32,35] that reported on the risk of long COVID in individuals who had received the primary course of vaccines versus unvaccinated individuals were included in this analysis (Fig. 4). Primary course vaccination was associated with a significantly lower risk of long COVID (p < 0.0001) compared with unvaccinated individuals, with a pooled OR of 0.81 (95% CI 0.79–0.83). Statistical heterogeneity was low ($I^2 = 0$%). However, sensitivity analyses could not be performed due to the small number of studies.

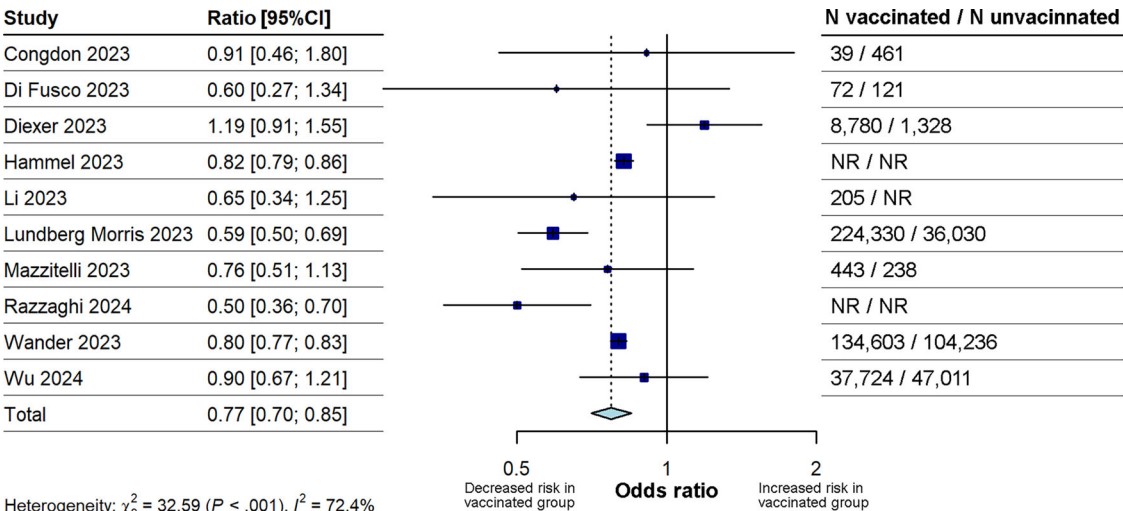

**Fig. 2 | Forest plot for the effect of any vaccination on the risk of long COVID compared with unvaccinated.** The figure illustrates pooled ORs and associated 95% CIs for the risk of long COVID after vaccination (any dose) compared with no vaccination. Random effects models were used, and all tests were 2-sided. Individual study estimates are shown as squares, with the size of the square proportional to study weight, and error bars indicating 95% CIs around the OR. The diamond represents the pooled effect size with its 95% CI, which was statistically significant (p < 0.0001). CI confidence interval, N total number, NR not reported, OR odds ratio.

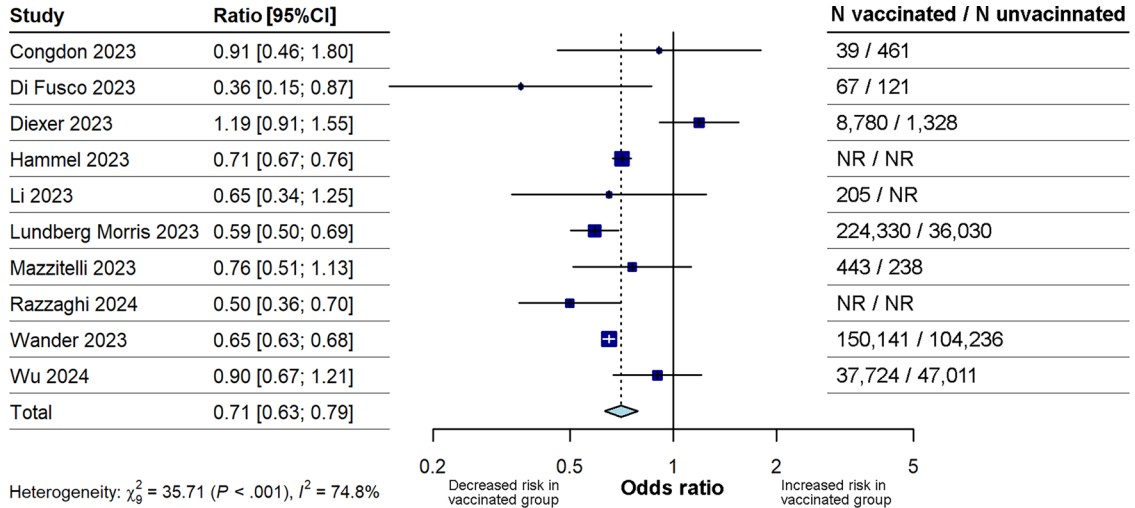

**Fig. 3 | Forest plot for the effect of "any vaccination" (analysis B) on the risk of long COVID compared with unvaccinated (sensitivity analysis substituting primary course estimates for booster dose estimates in three studies).** The figure illustrates pooled ORs and associated 95% CIs for the risk of long COVID after vaccination (any dose) compared with no vaccination for the sensitivity analysis that substitutes estimates for risk after primary course vaccination with estimates for risk after booster dose vaccination in three studies (Di Fusco 2023, Hammel 2023, Wander 2023). Random effects models were used, and all tests were 2-sided. Individual study estimates are shown as squares, with the size of the square proportional to study weight, and error bars indicating 95% CIs around the OR. The diamond represents the pooled effect size with its 95% CI, which was statistically significant (p < 0.0001). CI confidence interval, N total number, NR not reported, OR odds ratio.

**Impact of booster vaccination prior to Omicron variant infection on long COVID compared with no vaccination.** Four studies[10,32,35,46] that reported on the risk of long COVID in individuals who had received booster dose vaccination versus unvaccinated individuals were included in this analysis (Fig. 5).

Booster vaccination was associated with a significantly lowered risk of long COVID (p < 0.0001) compared with individuals who were unvaccinated, with a pooled OR of 0.74 (95% CI 0.63–0.86). Between study heterogeneity was high ($I^2$= 88.2%). Sensitivity analyses found that removal of either Wander 2023[35] or Hammel 2023[32] resulted in loss of statistical significance of the pooled OR value (Supplementary

Fig. 5). Removal of a potential overlapping population between studies also resulted in loss of significance of the pooled OR value (Supplementary Table 13).

**Impact of booster vaccination on long COVID compared with primary course vaccination prior to Omicron variant infection.** Three studies[10,46,52] that reported on the risk of long COVID in individuals who had received booster dose vaccination versus individuals who had received the primary course only were included in this analysis (Fig. 6). Booster vaccination was associated with a significantly lowered risk of long COVID (p = 0.004) compared with primary course vaccination

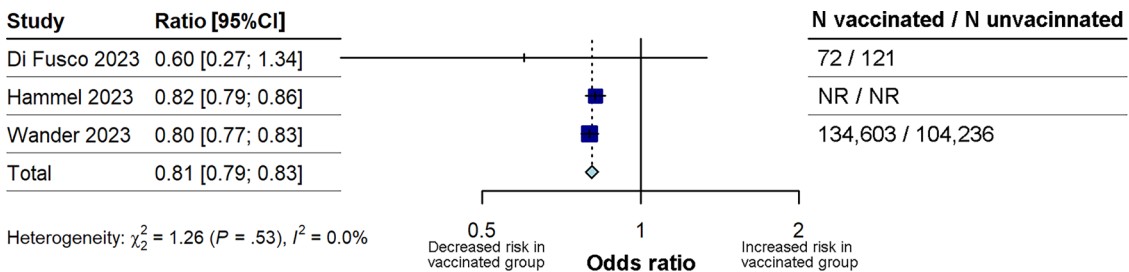

**Fig. 4 | Forest plot for the effect of primary course vaccination on the risk of long COVID compared with unvaccinated.** The figure illustrates pooled ORs and associated 95% CIs for the risk of long COVID after primary course vaccination compared with no vaccination. Random effects models were used, and all tests were 2-sided. Individual study estimates are shown as squares, with the size of the square proportional to study weight, and error bars indicating 95% CIs around the OR. The diamond represents the pooled effect size with its 95% CI, which was statistically significant (p < 0.0001). CI confidence interval, N total number, NR not reported, OR odds ratio.

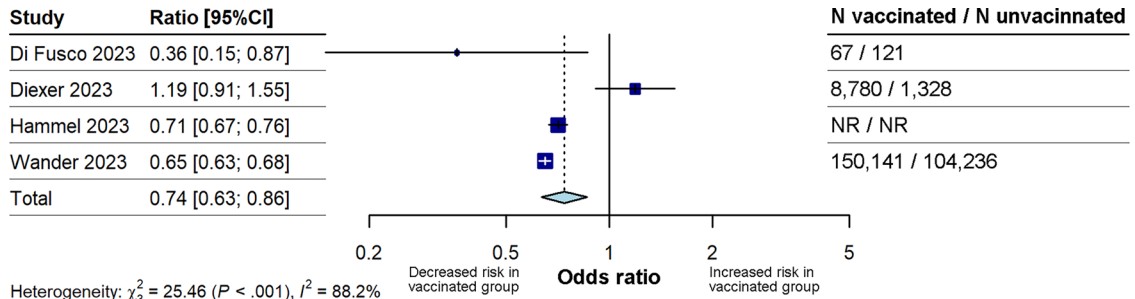

**Fig. 5 | Forest plot for the effect of booster vaccination on the risk of long COVID compared with unvaccinated.** The figure illustrates pooled ORs and associated 95% CIs for the risk of long COVID after booster vaccination compared with no vaccination. Random effects models were used, and all tests were 2-sided. Individual study estimates are shown as squares, with the size of the square proportional to study weight, and error bars indicating 95% CIs around the OR. The diamond represents the pooled effect size with its 95% CI, which was statistically significant (p = 0.0001). CI confidence interval, N total number, NR not reported, OR odds ratio.

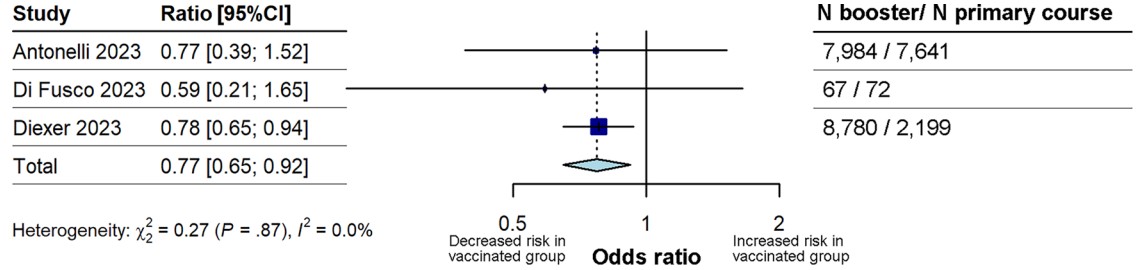

**Fig. 6 | Forest plot for the effect of booster vaccination on the risk of long COVID compared with primary course vaccination.** The figure illustrates pooled ORs and associated 95% CIs for the risk of long COVID after booster vaccination compared with primary course vaccination. Random effects models were used, and all tests were 2-sided. Individual study estimates are shown as squares, with the size of the square proportional to study weight, and error bars indicating 95% CIs around the OR. The diamond represents the pooled effect size with its 95% CI, which was statistically significant (p = 0.044). CI confidence interval, N total number, OR odds ratio.

only, with a pooled OR of 0.77 (95% CI 0.65–0.92). Between study heterogeneity was low ($I^2 = 0\%$). Sensitivity analyses could not be performed due to the small number of studies.

**Certainty of evidence.** The certainty of evidence for the outcomes assessed by GRADE was very low due to the observational design of the studies, serious inconsistency, and serious indirectness (Supplementary Table 14).

## Discussion

This systematic literature review and meta-analysis investigated the impact of COVID-19 vaccination, including booster doses, on preventing long COVID caused by subsequent Omicron variants

infections. The pooled OR of long COVID was 0.77 (95% CI 0.70–0.85) for participants receiving any vaccination compared with unvaccinated participants, while pooled ORs were 0.81 (95% CI 0.79–0.83) and 0.74 (95% CI 0.63–0.86) among those receiving primary course and booster vaccination, respectively, compared with unvaccinated participants. Among the vaccinated population, additional doses were associated with added protection against long COVID, demonstrated by a pooled OR of 0.77 (95% CI 0.65–0.92) for boosted versus primary course participants. This comparison between booster and primary course vaccination is critical, as it underscores the additional clinical advantages of seasonal COVID-19 vaccinations beyond the prevention of acute infections and severe disease.

Our analysis comparing primary vaccination versus no vaccination yielded results similar to those published in several pre-Omicron analyses[19,20,27], which reported significantly lower pooled risks of long COVID development of risk ratio 0.82 (95% CI 0.74–0.91)[20], OR 0.64 (95% CI 0.45–0.92)[27], and OR 0.54 (95% CI 0.30–0.99)[19]. For studies that included Omicron variants, in combination with non-Omicron variants, a significantly lower pooled risk of long COVID was reported for primary course vaccination (OR 0.65 [95% CI 0.62–0.68])[59]; for primary course and booster vaccination (OR 0.77 [95% CI 0.75–0.79])[60] and for booster doses only (OR 0.31 [95% CI 0.28–0.35])[23], all versus no vaccination. A previous meta-analysis performed a sub-analysis of Omicron infections only and reported a pooled risk of OR 0.68 (95% CI 0.54–0.86) for vaccinated versus unvaccinated participants[23], similar to the results presented here (OR 0.77 [95% CI 0.70–0.85]).

In summary, our findings support recent evidence suggesting that primary course vaccination and, to a greater extent, booster vaccination prior to SARS-CoV-2 infection may reduce the risk of long COVID symptoms. Further, a systematic literature review currently under peer-review has reported that existing data on COVID-19 vaccination in adults demonstrates that the effect of vaccination prior to infection on long COVID occurs regardless of the predominant variant in circulation[61].

## Factors affecting long COVID risk

A recent cohort study of 441,583 veterans from the Department of Veterans Affairs Health Care System databases by Xie et al. reported 5.23 fewer long COVID events per 100 persons at 1 year post infection during the Omicron era versus during the pre-Delta and Delta eras combined[4]. Furthermore, it was estimated that 28% of the reduction in long COVID events was attributable to viral era-related changes, with the remaining attributable to the effect of vaccination[4]. These findings demonstrate that the incidence of long COVID in the Omicron era remains substantial.

For the data presented here, it cannot be determined whether the protective effect of vaccination against long COVID arises solely from a reduced risk of infection and severe disease (population-level) or whether this can occur irrespective of acute disease severity. Severe COVID-19, as determined by hospitalization or intensive care unit admittance, is among the strongest predictors for the risk of developing long COVID[62,63]. It follows that, by effectively reducing the incidence of severe COVID-19, vaccination may, in turn, prevent progression to long COVID, thus achieving a protective effect. The impact of repeated SARS-CoV-2 infection on the risk of developing long COVID has not been fully elucidated. While some studies indicate that previous infection with SARS-CoV-2 may decrease the risk of long COVID after a second infection[46,64], there is also evidence to suggest that reinfection leads to a cumulative increased risk of long COVID[65,66].

There are a number of potential mechanisms by which vaccination may decrease the risk of long COVID. There is evidence to suggest long COVID could, in at least a subset of individuals, be attributed to persistence of SARS-CoV-2 RNA or protein, driving inflammatory, microbiome, and coagulation abnormalities[67,68]. Hence, a pre-existing vaccine-induced immune response could protect against the establishment of this so-called viral reservoir and, in turn, long COVID[67]. Findings have also suggested autoimmunity as a potential mechanism for long COVID pathogenesis[69], with a recent study indicating that transfer of immunoglobulin G antibodies from participants with long COVID can induce replicate symptoms in mice[70,71] as well as hyperinflammation caused by atherosclerotic plaque formation in cases of cardiovascular complications[13,72]; these processes could be mitigated by vaccination. Vaccination may also reduce the risk of long COVID by protecting against acute and post-acute inflammation that may lead to organ dysfunction or damage[73]. Studies have found that sustained viral burden and viral RNA load in the acute phase of COVID-19 are associated with an increased risk of long COVID, though the underlying mechanisms have not been determined[74–77]. It follows that vaccination could increase the speed of viral clearance and in turn mitigate downstream pathways for long COVID pathology.

## Implications for practice

Much of the initial messaging around COVID-19 vaccination was focused on the prevention of severe illness, hospitalization, and death. As the pandemic evolves and these outcomes become less common, understanding the effectiveness of vaccination against long COVID could help provide evidence for both the individual patient-level benefit and public health value of ongoing seasonal vaccination. This may be especially true for young, healthy individuals, who might otherwise consider themselves to be at low risk for acute complications of COVID-19. Findings on vaccine effectiveness, and more so the added benefit of boosters, could therefore inform guidelines on preventative medicine or primary care and assist healthcare professionals in providing evidence-based advice to their patients regarding vaccination. For public health, increasing seasonal vaccine uptake is an important element of the recently outlined global research and policy response strategy required to address the multifaceted challenges posed by long COVID[13]. Our analysis on booster versus primary vaccination provides evidence supporting COVID-19 seasonal vaccination policies.

## Quality of evidence

This systematic literature review and meta-analysis compared the impact of booster vaccination with primary course vaccination for Omicron-only infections, thus providing an improved understanding of the importance of seasonal vaccination against newer variants. In addition, there was a low or medium risk of bias in all but one[55] of the studies identified in the systematic literature review, and sensitivity analyses did not negate the statistical significance of the OR values, except in booster dose versus unvaccinated analyses.

It is important to note several limitations of this review. Only English-language studies were included, which may have limited the selection of data. Further, the statistical heterogeneity estimate ($I^2$) provided for the analyses should be interpreted with caution due to the limited number of studies[78]. The high $I^2$ of some analyses suggests that a large proportion of the observed variation in effect sizes across studies is due to true differences between the studies rather than just sampling error; we attempted to account for this with the use of a random effects model. As the analyses do not explore the effect of infection prevention since all participants had been infected with SARS-CoV-2 and presented with symptoms, the risk reductions described here are therefore conservative. There were limited subgroup analyses in the identified studies, meaning that sub-analyses could not be performed for sex, vaccine type, comorbidities, or acute infection severity. Most studies also did not adjust for acute illness severity, despite infection severity and hospitalization rate being reduced by vaccination[79]. Collider bias, a form of bias that can distort the exposure-outcome effect estimates in observational studies with non-representative samples[80], has important implications in COVID-19–related studies[81], especially after universal testing of mild cases no longer took place. However, collider bias was only explicitly stated to be considered by one of the included studies.

Data included in this meta-analysis were variable. The observational designs of the included studies introduced considerable heterogeneity to this review. In addition, there was significant variation in the definition of long COVID across studies[40,47,82] and those studies that use a broader definition of long COVID or self-assessment may overestimate or misdiagnose cases. The lack of consensus on the definition of long COVID remains an ongoing issue[83]. These variables were reflected in the GRADE analysis, where the certainty of evidence was very low.

Prevalence of long COVID has been reported to vary over time[84]; however, the follow-up periods of the included studies ranged from 4 weeks to 6 months depending on the definition of long COVID used. Although most definitions of long COVID have now converged on a minimum period of 3 months between infection and symptoms, we included the earlier definition of 1 or more month, as the shorter minimum time period was still in use during the period these studies were performed. Further, there were variations in vaccination status definitions across studies. For example, in one study, unvaccinated participants included those who received a monovalent dose ≥12 months before enrolment[53]. There is evidence that the effects of vaccination have not fully waned at 12 months after vaccination[85], so participants grouped using this definition may not be comparable to those who have never been vaccinated. Time between vaccination and Omicron infection was only reported by two of the studies included in the meta-analyis[10,22], preventing any sub-analyses to explore this factor, and risk outcomes were not adjusted for time between vaccination and infection by many studies. A limited number of studies in this review that assessed the impact of increasing time between vaccination and infection imply that vaccine effectiveness against long COVID may wane over time[22,31,52].

Another major limitation is the lack of randomization in the studies included in the meta-analysis and although studies adjusted for confounders, methods were not consistent across studies. Thus, the reduced risk of long COVID among patients who were vaccinated (or received a booster dose) may be due to systematic differences in the baseline characteristics of both groups.

### Implications for further research

As discussed, our meta-analysis is limited by varying definitions and heterogeneity in subgroup analyses. As such, an individual patient data meta-analysis could help address some of these drawbacks. Studies with appropriate sampling strategies to account for collider bias are also needed[81]. In addition, there are still many unresolved research questions regarding long COVID in the current Omicron era. For instance, the potential impact of seasonal vaccination on the risk of developing long COVID is unclear in people with hybrid immunity (repeat infection and seasonal vaccinations) compared with primary course vaccinated individuals. Given the success of global vaccination efforts and estimated seroprevalence, it will be difficult to recruit unvaccinated individuals or people who have not been exposed to SARS-CoV-2 infection, as a baseline comparator for future studies. Another important factor to investigate is the impact of combined flu and COVID-19 vaccination and vaccines for other diseases such as for respiratory syncytial virus (RSV) on the rates of long COVID, particularly as COVID-19 vaccinations have moved to a seasonal roll out (e.g. UK 2024 COVID-19 autumn vaccination[86]). Further, it is important to recognize that long COVID does not exist in isolation and thereby, gain an understanding of its intersection with broader healthcare challenges. A retrospective analysis of UK health records investigated these so called compound pressures and observed that flu vaccination and the preventative treatment for cardiovascular disease were associated with a reduced risk of hospitalization due to long COVID[87].

Previous studies have reported on the considerable healthcare costs associated with long COVID in the Omicron era[88–90]; however, it is important to investigate the cost-effectiveness of COVID-19 vaccines and update cost analyses, in relation to direct and indirect cost to healthcare systems and society, and as SARS-CoV-2 continues to evolve, to help inform government/public health policy.

In conclusion, both primary course and, to a greater extent, booster COVID-19 vaccination were significantly associated with a reduced risk of developing long COVID following Omicron variant infection. This cumulative protective effect against long COVID from additional booster doses is likely due to improved protection against severe acute COVID-19, known to be a key risk factor for the development of long COVID. However, these findings should be interpreted with consideration of the limitations associated with pooled observational real-world data. Our study highlights the importance of continued surveillance of SARS-CoV-2 variants in parallel to ongoing assessment of the effectiveness of current seasonal vaccines in preventing long COVID. These findings can inform evidence-based public health messaging around the individual and public health benefits of seasonal vaccination programs.

## Methods

This study was conducted in accordance with the Meta-analysis of Observational Studies in Epidemiology and the Preferred Reporting Items for Systematic Reviews and Meta-analyses (PRISMA) guidelines for conducting and reporting systematic reviews[91]. The study protocol was registered with PROSPERO (CRD42024501445).

### Data sources and searches

The following databases were used to identify relevant studies from 1 January 2022 to 1 March 2024 (date of searches): Embase, MEDLINE, PubMed, Europe PMC (including medRxiv and bioRxiv pre-print), Latin American and Caribbean Health Sciences Literature (LILACS), Cochrane COVID-19 Study Register, and the WHO COVID-19 Database. The start date for this search was selected to identify publications reporting on Omicron infections. Search strategies were developed using search terms related to long COVID and vaccination status. The complete search strategies are detailed in Supplementary Methods (1.1 Search Strategy). Hand searching was also performed, and bibliographies of identified studies were checked.

### Eligibility criteria, screening, and data extraction

Studies considered eligible for inclusion in the review met all the following criteria:

Population: participants (all ages) with Omicron variant SARS-CoV-2 infection. Infections were considered Omicron variants if studies either reported infections to be caused only by Omicron variants, included infections only occurring after November 2021, or reported infections that were only caused by B.1.1.529 and all subsequent subvariants arising from the Omicron lineage. Populations consisting of both Omicron and pre-Omicron variants were included only if outcomes of interest were reported separately for the Omicron population.

Intervention: participants who received COVID-19 vaccines, authorized by the European Union (EU) at the time of the study, prior to subsequent SARS-CoV-2 infection (BNT162b2 [Pfizer-BioNTech], mRNA-1273 [Moderna], Ad26.CoV2.S [Jannsen], ChAdOx1-S [AstraZeneca], Nuvaxovid [Novavax], VLA2001 [Valneva SE], or VidPrevtyn Beta [Sanofi-GSK]).

Comparison: either participants who were unvaccinated (for comparison with vaccinated) or received at least one fewer vaccine dose than the intervention population.

Outcomes: prevalence or incidence of long COVID (percentage of participants with symptoms, incidence rates), severity outcomes or symptom burden of long COVID disease (any measure of long COVID severity, average number of long COVID symptoms), and the risk of long COVID (e.g., OR, hazard ratio, risk ratio, rate ratio, incidence rate ratio, or vaccine effectiveness estimates). In this study, we defined long COVID as any condition or symptom described by the study as long COVID, PASC, PCC, or PACS, provided it was reported at least 4 weeks after the acute SARS-CoV-2 infection (based on the definition of long COVID used by the Centers for Disease Control and Prevention at the time of analysis[92]).

Study design: Observational (cohort, case-control, cross-sectional) studies.

Publication type and language: published and pre-print studies with full-texts available in English.

 

Titles and abstracts of identified references were assessed by two independent reviewers to determine whether they met the inclusion criteria. Any discrepancies were resolved by discussion until a consensus was reached. Full-text screening was performed on the studies that met the criteria for inclusion and data were extracted into a pre-defined table. One reviewer extracted data on study and participant characteristics, and outcomes of interest, and a second reviewer verified the accuracy of extraction. Data extracted are listed in Supplementary Methods 1.2 Data extraction.

## Quality assessment and GRADE assessment

The risk of bias of included studies was assessed using the NOS for cohort and case-control studies[93] (Supplementary Table 15), and by the JBI tool for cross-sectional studies[94]. Studies assessed by the NOS were categorized as low, medium, or high risk of bias according to their combined domain (selection, comparability, and outcome) scores, where the maximum score for each study can be 9. Studies were classed as low risk of bias if they scored 7–9 (7–8 for retrospective studies), medium risk if they scored 4–6, and high risk if they scored 0–3. Additionally, if a study did not score at least 1 in any of the three domains it was categorized as high risk.

The Grading of Recommendations Assessment, Development, and Evaluation (GRADE)[95] approach was used to assess the certainty of evidence for the impact of vaccination on the risk of long COVID. The GRADE approach involves assessing evidence across five domains: risk of bias, inconsistency, imprecision, and publication bias. Two researchers discussed the domains for each outcome until consensus was reached.

## Data analysis

Data synthesis was performed to identify key trends among the studies included in the systematic literature review. Study populations were divided into different vaccination status groups (unvaccinated, primary course vaccination, booster vaccination, and additional booster vaccination) according to the status and number of vaccine doses reported by the studies. Studies have frequently defined primary course vaccination as participants receiving two doses, while booster vaccination status has generally been defined as participants receiving ≥3 doses (additional booster doses reported by some studies have included a total of four or more vaccine doses). There has been an exception with the Ad26.COV2.S vaccine, where a single dose was considered a primary course and subsequent vaccination was considered a booster dose.

## Statistical analysis

A feasibility assessment was conducted among studies selected through the systematic literature review to determine whether pairwise meta-analyses could be performed for the risk of long COVID outcomes (Supplementary Methods: 1.3.1 Feasibility assessment). Pairwise meta-analyses were conducted to assess the risk of developing long COVID in those who had contracted the Omicron variant of SARS-CoV-2 after receiving a vaccine and/or booster doses compared with those who were unvaccinated or had received primary course vaccination.

If multiple timepoints were reported, the timepoint closest to that of other studies in the meta-analysis (3 months) was used for comparison. Most-adjusted analyses were selected for inclusion in the analyses. Hazard ratios and relative risk outcomes were converted into odds ratios (ORs) (Supplementary Methods: 1.3.2 Conversion of non-odds ratios to odds ratios). Vaccine effectiveness was converted to OR using $OR = 1 - (VE/100)$[96]. Ratios were log transformed, and weights were calculated using the inverse variance method (weight=1/variance). DerSimonian and Laird models (random effects)[97] were fitted to calculate pooled ORs and the 95% CI for all outcomes.

Sensitivity analyses were performed to assess the robustness of the results, and included leave-one-out analysis, substitution of risk in primary course versus unvaccinated outcomes with booster dose versus unvaccinated outcomes, removal of potentially overlapping populations or unadjusted results, and exclusion of adolescent/children-only studies, pre-print studies, high hospitalization for acute COVID illness studies (Supplementary Methods: 1.3.3 Sensitivity analyses). Heterogeneity was measured using the Cochran's Q test with statistical significance set as p < 0.05. The thresholds for interpretation of the I² statistic were defined by the Cochrane Handbook for Systematic Reviews of Interventions[58]. Publication bias was assessed using funnel plots and the Egger's test[98] for analyses that included at least 10 studies[58]. All statistical analyses were conducted using R version 4.1.1 with the meta package.

## Reporting summary

Further information on research design is available in the Nature Portfolio Reporting Summary linked to this article.

## Data availability

All summary data generated during this study are presented in this published article or the supplemental material provided. Original data can be found in the published articles which can be retrieved from publicly available databases.

## Code availability

The code used for the meta-analysis has been deposited in GitHub and can be accessed here: https://github.com/Maverex-Market-Access/Long-covid-vaccination-meta-analysis.

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

## Acknowledgements

The authors would like to thank Triantafyllos Pliakas (BioNTech SE and Impact Epilysis) for their contribution to the manuscript. Statistical support, including the design and running of the meta-analyses, was provided by Medha Shrivastava, MSc (Maverex Ltd). Medical writing support, including assisting authors with the development of the outline and initial draft, and incorporation of comments, was provided by Jasmine Bensilum, MSc, and editorial support was provided by Sarah Christopher, of Paragon (a division of Prime, Knutsford, UK), supported by BioNTech SE according to Good Publication Practice guidelines (Link). The sponsor was involved in the study design and collection, analysis, and interpretation of data, as well as data checking of information provided in the manuscript. However, ultimate responsibility for opinions, conclusions, and data interpretation lies with the authors. This study was funded by BioNTech SE.

## Author contributions

R.G., Z.M., G.Y.H.L., A.B., J.W., B.C.D., M.J.P., E.W., and S.A. contributed to conceptualization of the manuscript. R.G., Z.M., and B.D. contributed to methodology design. R.G. and Z.M. contributed to formal data analysis. R.G. and Z.M. contributed to the investigation process. R.G., Z.M., G.Y.H.L., A.B., J.W., B.C.D., M.J.P., E.W., and S.A. contributed to data analysis and interpretation. R.G., Z.M., G.Y.H.L., A.B., J.W., B.C.D., M.J.P., E.W., and S.A. contributed to the development of the original manuscript draft, as well as its subsequent review and editing. RG and Z.M. contributed to data visualization. B.D., A.B., and S.A. were responsible for research oversight and leadership. S.A. was responsible for funding acquisition and project administration.

## Competing interests

S.A. is an employee of BioNTech SE. R.G. and Z.M. are employees of Maverex Ltd both of whom received consulting fees from BioNTech SE. G.Y.H.L.: Consultant and speaker for BMS/Pfizer, Boehringer Ingelheim, Daiichi Sankyo, and Anthos. No fees were received personally. He is a National Institute for Health and Care Research (NIHR) Senior Investigator. M.J.P. has received consulting fees from Gilead Sciences, AstraZeneca, BioVie, Apellis Pharmaceuticals, and BioNTech and research support from Aerium Therapeutics and Shionogi, outside the submitted work. A.B.: Consultant for Perspectum. Speaker for Shionogi and Pfizer. J.W. has received honorarium from Banook, PPD, BioNTech, AMA, and Sanofi and research grants from Sanofi, Regeneron, and Boehringer Ingelheim. B.C.D.: BioNTech—one-off advisory board on long COVID in 2024. E.W. has received financial compensation from BioNTech.
