## [Peer Review file · Nature Communications]

A systematic review and meta-analysis of the impact of vaccination on prevention of long COVID

Corresponding Author: Dr Sultan Abduljawad

Version 0:

Reviewer comments:

Reviewer #1

(Remarks to the Author)

The study by Green et al. summarized information on vaccination and its effect on long COVID induced by SARS-CoV-2 Omicron variants. They concluded that primary and booster vaccination reduced the risk of long COVID during the Omicron era. Some additional information may be needed to help better understand and interpret the results of the study.

1. Please make it clear in the title and across the manuscript that the study measures the effect of vaccination before infection within those infected.
2. Multiple factors could affect vaccine effectiveness, including the time between the last vaccination and infection. It is common that those who received booster vaccines with less time between the last vaccination and infection. Could the authors clarify if this has been considered? In addition, please provide the time intervals between the last vaccination and infection for the different studies included.
3. Given the varied effects of Omicron subvariants on long COVID and the varied immune response profiles for different subvariants, I suggest the authors list the dominant or predominant variants for each study. While patient level data may not be always available, this could be proxied by area/national-level data.
4. The composition of the booster examined in the study is not clear. Does it refer only to one additional dose of a monovalent vaccine targeting the original strain on top of the primary vaccination? Or does it include the Omicron-adapted bivalent or monovalent vaccines?
5. Please provide 95% confidence intervals when mentioning vaccine effectiveness.

(Remarks on code availability)

Reviewer #2

(Remarks to the Author)

This systematic review and meta-analysis investigating the impact of COVID-19 vaccination on preventing long COVID in the Omicron era. Several aspects require attention and clarification.

Introduction

What is SLR?

Methods:
Search Strategy

Justify the date range selection (January 2022 to March 2024)

Study Selection and data extraction

The inclusion/exclusion criteria for "EU-authorized vaccines" needs more precise definition

Explain how studies with mixed variants were handled, particularly those that included both Omicron and other variants

Clarify how overlapping populations between studies were identified and managed

Please provide details for extracted data

Data Analysis

How to define unvaccinated, primary course vaccination, booster vaccination, and additional booster vaccination?

Statistical analysis

Considering OR, HR, and RR as equal estimates is inappropriate.

How to deal with adjusted and unadjusted results?

Consider conducting additional sensitivity analyses based on study quality.

It is important to assess the certainty of the evidence using tools such as GRADE.

Results:

Study Characteristics

Provide more details about the vaccination protocols (e.g., timing between doses, specific vaccine combinations)

Discussion:

Address the impact of vaccination timing relative to infection

Consider discussing the implications of waning immunity

Limitations

Discuss the impact of varying follow-up periods

Address the limitation of not being able to control for disease severity

Explain the potential impact of different long COVID definitions on the pooled estimates

Address the high heterogeneity ($I^2 > 70\%$) in some analyses and its implications.

(Remarks on code availability)

Reviewer #3

(Remarks to the Author)

Dear [Editor/Authors],

Thank you for the opportunity to review the interesting work by Green et al.

Long COVID, defined as persistent symptoms following SARS-CoV-2 infection, is typically characterized by symptoms lasting at least three months after the initial infection^{1,2}. While this three-month definition is commonly used in studies, variations in definitions are evident across the research community. Moreover, it is important to acknowledge that similar post-viral conditions may occur after infections with other respiratory viruses^{3,4}. In the current situation, with high levels of immunity against SARS-CoV-2, our understanding of long COVID in comparison to long-term symptoms induced by other infections remains limited. For example, studies comparing risk of long-term symptoms after SARS-CoV-2 infection and influenza have reported mixed results^{3,4}.

In estimating vaccine effectiveness against long COVID, several methodological challenges in observational studies must be considered. These include the difficulty of addressing confounding, which may arise from differences in testing practices, healthcare-seeking behaviours, and the presence of chronic conditions across exposed and unexposed. Additionally, diagnosing long COVID itself is complex. Furthermore, studies comparing the risk of long COVID between vaccinated and unvaccinated individuals are particularly vulnerable to collider bias^{5,6}.

In this meta-analysis, Green et al. observed that COVID-19 vaccination was associated with a modest reduction (approximately 20%) in the risk of long COVID symptoms among individuals infected with the Omicron variant. Previously, some meta-analysis have been already conducted limiting the impact of this meta-analysis. Moreover, the current interest would be the benefit of variant-adapted COVID-19 vaccines as a seasonal booster for long-covid symptoms. Still, the review is well-conducted, and the manuscript is clear and comprehensible. Below, I provide my detailed comments for the authors:

Major Comments

Highlighting Limitations of Observational Studies

The manuscript briefly mentions potential residual confounding and biases in the discussion, but these limitations should be

emphasized more prominently, particularly in the introduction. This would improve the objectivity of how the findings are presented.

Collider Bias in Omicron-Infected Populations (Lines 120–123)

If the analysis is limited to individuals already infected with Omicron, this introduces a serious concern of collider bias⁶. For instance, the severity of Omicron infection may vary across vaccinated and unvaccinated groups, potentially skewing results. Moreover, other sources for collider bias should be also considered. The authors should address how the problem of collider bias was managed and consider discussing additional mechanisms by which collider bias might operate in the studies.

Definition of Long COVID (Lines 137–138)

The most widely accepted definition of long COVID requires symptoms to persist for at least three months. Why was this standard definition not used in the analysis? Could a sensitivity analysis using the three-month criterion be included to ensure comparability with the current golden standard definition?

Time Since Vaccination (Lines 125–127)

Time since vaccination - meaning time from vaccination until Omicron infection - should be considered as a key factor in the analysis. Vaccine effectiveness may diminish over time, and studies with shorter intervals between vaccination and infection likely observe greater effectiveness. Without this consideration, the applicability of the results is limited.

COVID-19 vaccines

What were the most commonly used vaccines in the studies? This is not well described in Supplementary Table 3. Additionally, the manuscript should include some mention of the most commonly used vaccines.

Minor Comments

Clarification of Observational Studies (Lines 76–77)

Please clarify that studies 14–21 are observational by explicitly stating, for example: “Vaccination prior to SARS-CoV-2 infection has been associated with a reduced risk of developing long COVID in observational studies.”

Analysis A and B in Methods Section

Analyses A and B are not sufficiently described in the methods section. Please ensure these are clearly explained to enhance transparency and reproducibility.

Sensitivity Analyses (Lines 182–183)

The manuscript refers to sensitivity analyses, but these are not clearly described in the methods section. Please specify the methods and results of these analyses in the appropriate section.

Collider Bias in Studies 73–74 (Lines 368–372)

Studies 73–74^{7,8} are likely influenced by collider bias, as reinfected individuals are selected based on specific characteristics. These include:

- Immunocompromised status or other conditions that increase the likelihood of reinfection. The proportion of these conditions will be higher among reinfected individuals, as previous infection provides higher protection against reinfection in immunocompetent and otherwise healthy individuals.
- Variability in first infection vs reinfection identification, with only severe reinfections being more likely to be tested and documented.

Additionally, the works of Bowe et al.⁷ and Barboza et al.⁸ contradict the findings of this meta-analysis. SARS-CoV-2 infection induces an immune response comparable to, or even stronger than, COVID-19 vaccination. Could the authors clarify the immunological mechanisms that reduce long COVID risk after vaccination but increase it following prior infection?

References

1. Post COVID-19 condition (Long COVID). <https://www.who.int/europe/news-room/fact-sheets/item/post-covid-19-condition>.
2. CDC. Long COVID Basics. COVID-19 <https://www.cdc.gov/covid/long-term-effects/index.html> (2024).
3. Xie, Y., Choi, T. & Al-Aly, Z. Long-term outcomes following hospital admission for COVID-19 versus seasonal influenza: a cohort study. *The Lancet Infectious Diseases* 24, 239–255 (2024).
4. Brown, M. et al. Ongoing symptoms and functional impairment 12 weeks after testing positive for SARS-CoV-2 or influenza in Australia: an observational cohort study. *bmjph* 1, e000060 (2023).
5. Hernán, M. A. & Monge, S. Selection bias due to conditioning on a collider. *BMJ* p1135 (2023) doi:10.1136/bmj.p1135.
6. Holmberg, M. J. & Andersen, L. W. Collider Bias. *JAMA* 327, 1282 (2022).
7. Bowe, B., Xie, Y. & Al-Aly, Z. Acute and postacute sequelae associated with SARS-CoV-2 reinfection. *Nat Med* 28, 2398–2405 (2022).

8. Barboza, A. P. et al. Long COVID in a Multicentric Brazilian Cohort: Acute Phase Symptom Burden and Second Booster Dose Vaccination Effects. Preprint at <https://doi.org/10.2139/ssrn.4571726> (2023).

(Remarks on code availability)

The code seems appropriate

Version 1:

Reviewer comments:

Reviewer #2

(Remarks to the Author)

Despite the authors' revisions, serious methodological flaws persist, critically undermining the study's validity.

1. Treating odds ratios (ORs), hazard ratios (HRs), and risk ratios (RRs) as interchangeable effect measures constitutes a fundamental methodological flaw that critically undermines the validity of this meta-analysis. These indices quantify distinct epidemiological relationships—ORs disproportionately amplify effects for common outcomes (>10%), while HRs incorporate temporal dynamics absent in RRs. Pooling them without conversion introduces severe clinical and statistical heterogeneity, likely distorting the pooled effect size and rendering clinical interpretations unreliable. To rectify this, we strongly demand that the authors apply validated conversion methods (e.g., Grant RL. Converting an odds ratio to a range of plausible relative risks for better communication of research findings. *BMJ* 2014; 348: f7450. Wang Z. Converting odds ratio to relative risk in cohort studies with partial data information. *Journal of Statistical Software* 2013; 55(5): 1-11.) to harmonize all estimates into clinically interpretable RRs, followed by reanalysis using the unified metrics. This essential correction aligns with Cochrane and PRISMA standards for evidence synthesis and is non-negotiable for ensuring credible, clinically applicable conclusions.

2. In observational studies, many confounding factors can influence the outcomes, but the authors included only one study that adjusted for confounders in its results, while the majority of the included studies did not adjust for confounding factors. Therefore, the findings of this meta-analysis are likely unreliable and could even mislead clinical practice.

3. While we acknowledge concerns regarding heterogeneity and the review's exploratory intent, the application of GRADE remains imperative to uphold methodological rigor. GRADE is expressly designed to evaluate evidence certainty despite limitations like study diversity—its structured framework transparently quantifies how factors such as inconsistency (heterogeneity), risk of bias, and indirectness impact confidence in findings. Omitting GRADE undermines the review's validity, as readers cannot discern whether observed patterns reflect reliable associations or methodological artifacts. Even for non-clinical syntheses, certainty grading is essential to: (1) contextualize "exploratory" results, (2) prioritize future research needs, and (3) prevent overinterpretation of unstable data. We strongly recommend authors to use GRADE approach to assess the certainty of evidence.

Reviewer #3

(Remarks to the Author)

Thank you for your clear response! I don't have any further comments. Good luck!

Reviewer #4

(Remarks to the Author)

All of the responses related to reviewer no. 1 previous comments were answered by the authors. All of the responses were satisfactory and have addressed the raised concern

Version 2:

Reviewer comments:

Reviewer #2

(Remarks to the Author)

Thank you for your response! I don't have any further comments

Response to reviewers' comments

Reviewer 1

Please make it clear in the title and across the manuscript that the study measures the effect of vaccination before infection within those infected.

Thank you for this suggestion – we have now clarified that we are assessing prevention of long COVID in subsequent infections following vaccination in the Abstract (lines 27–28), Introduction (line 105), and Discussion (line 343). However, due to character limits for the title, we are unable to include this there.

Multiple factors could affect vaccine effectiveness, including the time between the last vaccination and infection. It is common that those who received booster vaccines with less time between the last vaccination and infection. Could the authors clarify if this has been considered? In addition, please provide the time intervals between the last vaccination and infection for the different studies included.

We agree with the reviewer that time between vaccination and infection may influence the impact of vaccination on preventing long COVID. However, this information is not provided by most of the studies (and by two that do, this information is reported for the entire population of mixed variants, not for Omicron-only), limiting the extent to which we can evaluate the effect of this time interval on vaccine effectiveness. Nonetheless, we have now expanded Supplementary Table 3 to now present any available time measurement between the last vaccine dose and SARS-CoV-2 infection and included this point in the Results: Vaccination Status (lines 268–270). We have also discussed this limitation in the Discussion (lines 471–477).

Given the varied effects of Omicron subvariants on long COVID and the varied immune response profiles for different subvariants, I suggest the authors list the dominant or predominant variants for each study. While patient level data may not be always available, this could be proxied by area/national-level data.

Thank you for this suggestion – we have now included the subvariants reported by the studies in Table 1 (Characteristics of studies included in the SLR). For those studies that did not report subvariant information, we considered the Omicron infection period reported by each study – most studies included infections that occurred when the BA.1/2 or BA.4/5 subvariants were dominant. We have now included this information in Results: Overview of included studies (lines 238–240). As not all studies clearly reported end dates for the infection period, we chose not to use area/national-level data as a proxy as we cannot be certain of the accuracy of our estimations.

The composition of the booster examined in the study is not clear. Does it refer only to one additional dose of a monovalent vaccine targeting the original strain on top of the primary vaccination? Or does it include the Omicron-adapted bivalent or monovalent vaccines?

Supplementary Table 3 now provides all details regarding doses where this information was reported by the included studies. Additionally, we have included clarification in Results: Overview of included studies (lines 253–254) that states that only a single study reported that participants received a bivalent booster vaccine.

Please provide 95% confidence intervals when mentioning vaccine effectiveness.

95% confidence intervals have been added to the relevant data in both the Abstract (lines 31–36) and the Discussion (lines 344–351).

Reviewer 2

What is SLR?

Thank you for spotting this – this is now defined at the first use of this abbreviation (line 99).

Search Strategy: Justify the date range selection (January 2022 to March 2024)

The reasoning for the start date of 1 January 2022 is now stated in the Methods: Data sources and searches section (line 117–118); this date was selected so studies reporting on Omicron variants would be identified, and studies published prior to this would only include earlier variants. We have also clarified that the end date for searches (1 March 2024) was the date the searches were performed.

Study Selection and data extraction:

- **The inclusion/exclusion criteria for "EU-authorized vaccines" needs more precise definition**
The Intervention inclusion criteria now comprehensively lists all vaccines (and the manufacturer) authorised by the EMA during the time period this review covers, regardless of whether they were later withdrawn (lines 132–136).
- **Explain how studies with mixed variants were handled, particularly those that included both Omicron and other variants**
 - The Population inclusion criteria has been expanded to clarify this: “Populations consisting of both Omicron and pre-Omicron variants were included only if outcomes of interest were reported separately for the Omicron population” (lines 129–131)
- **Clarify how overlapping populations between studies were identified and managed**
 - We have now clarified how potential overlapping populations were identified in the Supplementary Methods 1.3.2 Sensitivity analyses: “Data sources for all studies were compared; participant inclusion characteristic and time period for those studies with the same data sources were then compared to determine whether it was possible for any participants to be included in both studies.” (lines 584–588)
- **Please provide details for extracted data**
 - We have now included a list of the data extracted from studies – this is presented in Supplementary Methods 1.2 Data extraction (lines 557–564) due to word count limits.

Data analysis: How to define unvaccinated, primary course vaccination, booster vaccination, and additional booster vaccination?

This information was provided in the Results, however upon consideration of the reviewer's comment, we agree that this information should be provided earlier in the Methods section. We have therefore moved this information to the Methods (lines 176–182).

Statistical analyses:

- **Considering OR, HR, and RR as equal estimates is inappropriate.**
 - We made the decision to log transform OR, RR, and HR and use a random-effects model to account for potential differences in effect measures, as this can be considered possible under some circumstances such as when approximate equivalence can be assumed such as in cases where the event is rare (Greenland 1987; <https://doi.org/10.1093/oxfordjournals.epirev.a036298>). This method has been used previously in other similar COVID-19 meta-analyses (Parohan 2020; <https://doi.org/10.1080/13685538.2020.1774748>).
- **How to deal with adjusted and unadjusted results?**
 - This is a good point by the reviewer – all but one of the studies included in the meta-analyses had adjusted results. We have therefore included an additional sensitivity analysis that removes this unadjusted result (Supplementary Tables 10 and 11). Additionally, we have clarified in the Methods: Statistical analysis that the most-adjusted results were included in analyses when multiple adjustments were performed (lines 197–198).
- **Consider conducting additional sensitivity analyses based on study quality.**
 - As all studies included in the meta-analyses were considered low or medium risk of bias (Supplementary Table 2), it was not necessary to perform sensitivity analyses removing high risk of bias studies. We have now clarified this in the Supplementary Results 2.1 Meta-analyses feasibility assessment.
- **It is important to assess the certainty of the evidence using tools such as GRADE.**
 - While GRADE can be applied to observational studies, we did not consider it appropriate for this review, due mainly to the high heterogeneity of the included studies in terms of study design, populations, and outcome, which would make any certainty assessments unreliable. Additionally, GRADE was not performed as this review is not intended to inform clinical decisions directly.

Results: Provide more details about the vaccination protocols (e.g., timing between doses, specific vaccine combinations)

We have expanded Supplementary Table 3, which now includes all details of vaccination reported by the included studies. This table now includes:

- The number of doses (and the number/percentage of Omicron-infected study participants who received them)
- The minimum time required between last vaccine dose and SARS-CoV-2 infection for participants to be considered vaccinated before infection
- Any measurement of time between last vaccine dose and SARS-CoV-2 infection
- Vaccine type received (including number/percentages where reported)
- Timing between doses were considered, however this was not reported by any of the included studies.

Discussion: Address the impact of vaccination timing relative to infection. Consider discussing the implications of waning immunity

We have now addressed the impact of time between vaccination and infection in the Discussion (lines 471–477), in the limitations section as most studies did not report this information or adjust for this confounder. Although the impact of increasing time between vaccination and infection was not explored by most studies, we have also discussed the findings of those few studies that did.

Limitations:

- **Discuss the impact of varying follow-up periods**
- **Address the limitation of not being able to control for disease severity**
- **Explain the potential impact of different long COVID definitions on the pooled estimates**
- **Address the high heterogeneity ($I^2 > 70\%$) in some analyses and its implications**

We have now expanded the limitations section to include greater discussion on these four limitations

- Follow-up periods: (lines 460–461)
- Disease severity: (lines 448–450)
- Long COVID definition: (lines 457–458)
- High heterogeneity: (lines 440–443)

Reviewer 3

MAJOR COMMENTS

Highlighting Limitations of Observational Studies. The manuscript briefly mentions potential residual confounding and biases in the discussion, but these limitations should be emphasized more prominently, particularly in the introduction. This would improve the objectivity of how the findings are presented.

Thank you for this suggestion – we have now expanded the discussion of limitations and biases of the included studies due to their observational nature in the Discussion: Quality of evidence section (lines 455–456).

Collider Bias in Omicron-Infected Populations (Lines 120–123). If the analysis is limited to individuals already infected with Omicron, this introduces a serious concern of collider bias⁶. For instance, the severity of Omicron infection may vary across vaccinated and unvaccinated groups, potentially skewing results. Moreover, other sources for collider bias should be also considered. The authors should address how the problem of collider bias was managed and consider discussing additional mechanisms by which collider bias might operate in the studies.

We would like to thank the reviewer for this very thoughtful comment. We acknowledge that collider bias can distort the exposure-outcome effect estimates in observational studies with non-representative samples. The use of appropriate sampling strategies at the study design stage is the most efficient way to address collider bias (Griffith et al 2020;

<https://doi.org/10.1038/s41467-020-19478-2>). In our review, only one of the included studies considered collider bias (Wu 2024).

Rijnhart et al (2024) highlighted the importance of considering collider bias, in addition to confounding bias and overadjustment bias, in guidelines and tools for systematic literature review and meta-analyses of observational studies (Rijnhart et al 2024; <https://doi.org/10.1093/ije/dyae147>). We acknowledge that the PRISMA guidelines we used in our study do not provide relevant guidance on adjustment for colliders (Rijnhart et al 2024). To address this, we have included a discussion on collider bias in the limitations (line 450–454) and have provided additional discussion on the severity of infection, which was poorly reported in the studies that were included in the meta-analysis (line 448–450).

Definition of Long COVID (Lines 137–138). The most widely accepted definition of long COVID requires symptoms to persist for at least three months. Why was this standard definition not used in the analysis? Could a sensitivity analysis using the three-month criterion be included to ensure comparability with the current golden standard definition?

The definition of long COVID as symptoms persisting for at least 4 weeks after the acute illness was used in this review as at the time the included studies were performed, this definition was still being used by some organisations including the FDA and NICE. Although the three-month definition is now accepted, by using this newer definition of long COVID we would be excluding studies that adhered to an accepted definition of long COVID at the time – we have now included this point in the Discussion (lines 462–466). However, where studies reported multiple outcomes for different timepoints (2 studies, reporting outcomes for 1 month and 3 months), only the 3-month outcome was included in the meta-analyses as this timepoint was closest to that of the other studies (this is mentioned in the Methods: statistical analyses) (lines 196–197)

Time Since Vaccination (Lines 125–127). Time since vaccination - meaning time from vaccination until Omicron infection - should be considered as a key factor in the analysis. Vaccine effectiveness may diminish over time, and studies with shorter intervals between vaccination and infection likely observe greater effectiveness. Without this consideration, the applicability of the results is limited.

We agree with the reviewer that time since vaccination is an important factor to consider. We have now included these data where reported by the included studies in Supplementary Table 3; however, only six studies included this information (two of which only reported time for the entire mixed-variant population) and therefore this review cannot explore the impact of time. However, we have included this point in the Limitations section of the Discussion, as well as discussed the findings of those few studies that compared vaccination effectiveness against long COVID between groups with different time periods between vaccination and infection (lines 471–477).

COVID-19 vaccines. What were the most commonly used vaccines in the studies? This is not well described in Supplementary Table 3. Additionally, the manuscript should include some mention of the most commonly used vaccines.

Supplementary Table 3 now includes full details of the different vaccines used in the included studies (please note that for most studies that included multiple vaccine types, most studies did not report the number/percentages of participants receiving each type). We have also included a sentence in Results: Overview of included studies stating that mRNA vaccines were the most frequently reported vaccines included in the studies (lines 245–246).

MINOR COMMENTS

Clarification of Observational Studies (Lines 76–77). Please clarify that studies 14–21 are observational by explicitly stating, for example: “Vaccination prior to SARS-CoV-2 infection has been associated with a reduced risk of developing long COVID in observational studies.”

Thank you for the suggestion – we have now clarified this by stating these are observational studies (line 78).

Analysis A and B in Methods Section. Analyses A and B are not sufficiently described in the methods section. Please ensure these are clearly explained to enhance transparency and reproducibility.

We agree with the reviewer that the description of Analysis A and B needs to be included in the Methods – we have now included a description of it in the Methods: Statistical analysis (lines 191–195) section that we hope explains this more clearly.

Sensitivity Analyses (Lines 182–183). The manuscript refers to sensitivity analyses, but these are not clearly described in the methods section. Please specify the methods and results of these analyses in the appropriate section.

Thank you for pointing this out – we have now listed the sensitivity analyses performed in the Methods section (lines 204–207) and expanded the full descriptions of these analyses in the Supplementary Methods (1.3.2 Sensitivity analyses).

Collider Bias in Studies 73–74 (Lines 368–372). Studies 73–74,78 are likely influenced by collider bias, as reinfected individuals are selected based on specific characteristics. These include:

- Immunocompromised status or other conditions that increase the likelihood of reinfection. The proportion of these conditions will be higher among reinfected individuals, as previous infection provides higher protection against reinfection in immunocompetent and otherwise healthy individuals.
- Variability in first infection vs reinfection identification, with only severe reinfections being more likely to be tested and documented.

Additionally, the works of Bowe et al. and Barboza et al. contradict the findings of this meta-analysis. SARS-CoV-2 infection induces an immune response comparable to, or even stronger than, COVID-19 vaccination. Could the authors clarify the immunological mechanisms that reduce long COVID risk after vaccination but increase it following prior infection?

We agree with the reviewer that these two studies mentioned in the Discussion are likely influenced by collider bias due to the reasons listed. These studies were not included in our systematic review and meta-analysis as they did not meet our inclusion criteria.

Regarding the immunological mechanisms that reduce long COVID risk after vaccination but increase it following prior infection, this is an interesting question – while a specific immunological mechanism has not yet been identified, infections can however have negative downstream effects on different biological systems and can cause disease with varying severity. There are a number of mechanisms that could be exclusive to infection. For example, viral persistence might be expected after infection rather than vaccination, while other mechanisms such as autoimmunity or Epstein-Barr virus reactivation may be more likely to be set off by infection rather than vaccination. Although vaccination does induce an inflammatory response, only induces immunity to certain components of the virus and so cannot cause COVID-19 disease,

which is the reason for developing long COVID. We have discussed potential mechanisms by which vaccination may decrease the risk of long COVID (lines 395–412), however as this review did not investigate the impact of prior infection, we chose not to expand this to prior infections mechanisms.

- 1. Treating odds ratios (ORs), hazard ratios (HRs), and risk ratios (RRs) as interchangeable effect measures constitutes a fundamental methodological flaw that critically undermines the validity of this meta-analysis. These indices quantify distinct epidemiological relationships—ORs disproportionately amplify effects for common outcomes (>10%), while HRs incorporate temporal dynamics absent in RRs. Pooling them without conversion introduces severe clinical and statistical heterogeneity, likely distorting the pooled effect size and rendering clinical interpretations unreliable. To rectify this, we strongly demand that the authors apply validated conversion methods (e.g., Grant RL. Converting an odds ratio to a range of plausible relative risks for better communication of research findings. *BMJ* 2014; 348: f7450. Wang Z. Converting odds ratio to relative risk in cohort studies with partial data information. *Journal of Statistical Software* 2013; 55(5): 1-11.) to harmonize all estimates into clinically interpretable RRs, followed by reanalysis using the unified metrics. This essential correction aligns with Cochrane and PRISMA standards for evidence synthesis and is non-negotiable for ensuring credible, clinically applicable conclusions.**

We have carefully considered the reviewer's recommendation for conversion of ratios and have performed a conversion of ratios that we believe addresses the concerns raised.

We initially assessed the feasibility of converting all odds ratios (OR; 6 studies) and hazard ratios (HR; 3 studies) into relative risk (RR) ratios, as outlined in the references the reviewer suggested. However, as most of the OR-reporting studies did not report the data required to perform this conversion (i.e., risk of long COVID in the comparator group, or the number of patients this can be calculated from), we were therefore unable to do this.

It was possible to convert the non-odds ratios to odds ratios given they a) have a low (<10%) occurrence rate (i.e., long COVID development) and b) a relatively short time period (follow-up period after COVID infection) (Symons 2002, Wang 2013). We have presented the equations used in these conversions in the supplementary materials (section 1.3.2: *Conversion of non-odds ratios to odds ratios*). We have also confirmed that these methods have been used in previously published meta-analyses (e.g., Shor 2017).

The conversions of these non-odds ratios resulted in odds ratios with highly similar values to the original HR/RR (a maximum change of 0.02), which are all presented in the table below. This low divergence is expected, due to the short length of follow-up, the low endpoint occurrence over the follow-up period, and the magnitude of risk below 1 (Symons 2002). Due to these reasons, we do not see amplification (severe statistical heterogeneity) in OR conversions.

Study	Original	Converted to OR
Hammel 2023 (primary course vs no vaccination)	HR 0.82 (0.79, 0.86)	0.82 (0.79, 0.86)
Hammel 2023 (booster dose vs no vaccination)	HR 0.72 (0.68, 0.76)	0.71 (0.67, 0.76)
Lundberg-Morris 2023 (vaccinated vs no vaccination)	HR 0.59 (0.50, 0.69)	0.59 (0.50, 0.69)
Wander 2023 (primary course vs no vaccination)	HR 0.80 (0.78, 0.83)	0.80 (0.77, 0.83)
Wander 2023 (booster dose vs no vaccination)	HR 0.66 (0.64, 0.69)	0.65 (0.63, 0.68)
Wu 2024 (vaccinated vs unvaccinated)	RR 0.91 (0.69, 1.19)	0.90 (0.67, 1.21)

After re-running the meta-analyses with these new ratios, the pooled ORs showed extremely small divergence from the original analyses (p values for significance were also unchanged):

Analysis	New analysis	Original
Vaccinated vs unvaccinated (analysis A):	0.77 (0.70, 0.85)	0.78 (0.71, 0.85)
Vaccinated vs unvaccinated (analysis B):	0.71 (0.63, 0.79)	0.71 (0.64, 0.80)
Primary course vs unvaccinated:	0.81 (0.79, 0.83)	0.81 (0.79, 0.83)
Booster dose vs unvaccinated: 0.81 (0.79, 0.83)	0.74, 0.63, 0.86)	0.74, 0.65, 0.86)

There was no change to the “booster dose vs primary course” analysis as all original measures were OR.

We have therefore made the following changes to the manuscript:

- Methods: Replaced the sentence about all ratios considered equal estimates with “Hazard ratios and relative risk outcomes were converted into odds ratios (ORs) (Supplementary Methods: 1.3.2 Conversion of non-odds ratios to odds ratios)” (lines 204–206)
- Supplementary: Included a new section (1.3.2: *Conversion of non-odds ratios to odds ratios*) that explains the justification for converting non-ORs to ORs, and the equations used to perform these conversions. For full transparency, this section also includes the original ratios, and the converted ratios to RR and ORs, as well as the P_0 used in the conversion for each ratio
- All text, figures, and tables throughout have been updated with the revised pooled risks, including the sensitivity analyses and Eggers’ test results in the supplementary materials

Symons, M. J., & Moore, D. T. (2002). Hazard rate ratio and prospective epidemiological studies. *Journal of clinical epidemiology*, 55(9), 893-899.

Wang, Z. (2013). Converting odds ratio to relative risk in cohort studies with partial data information. *Journal of Statistical Software*, 55, 1-11.

Shor, E., Roelfs, D., & Vang, Z. M. (2017). The “Hispanic mortality paradox” revisited: meta-analysis and meta-regression of life-course differentials in Latin American and Caribbean immigrants' mortality. *Social Science & Medicine*, 186, 20-33.

- 2. In observational studies, many confounding factors can influence the outcomes, but the authors included only one study that adjusted for confounders in its results, while the majority of the included studies did not adjust for confounding factors. Therefore, the findings of this meta-analysis are likely unreliable and could even mislead clinical practice.**

We apologise for the confusion, but all of the included studies performed adjustments for potential confounding factors except for one (a sensitivity analysis was performed removing this single unadjusted result which showed no significant change in P value). We believe this confusion arises from our statement that only one study considered collider bias; we have now rephrased this

sentence to make it clear that only one study explicitly stated that they attempted to address the risk of collider bias in their study (lines 465-466).

- 3. While we acknowledge concerns regarding heterogeneity and the review's exploratory intent, the application of GRADE remains imperative to uphold methodological rigor. GRADE is expressly designed to evaluate evidence certainty despite limitations like study diversity—its structured framework transparently quantifies how factors such as inconsistency (heterogeneity), risk of bias, and indirectness impact confidence in findings. Omitting GRADE undermines the review's validity, as readers cannot discern whether observed patterns reflect reliable associations or methodological artifacts. Even for non-clinical syntheses, certainty grading is essential to: (1) contextualize "exploratory" results, (2) prioritize future research needs, and (3) prevent overinterpretation of unstable data. We strongly recommend authors to use GRADE approach to assess the certainty of evidence.**

We acknowledge the reviewer's concerns about omitting GRADE, and thank them for their thorough reasoning. We have therefore performed GRADE analysis for this meta-analysis and included the following in the revised manuscript:

- GRADE methods (lines 171–176)
- Results: Certainty of evidence section summarising the results of the assessment (lines 349–351)
- Discussion: The very low certainty of evidence score is mentioned in the Limitations section (lines 472–473)
- Conclusions: To address the concerns about the reliability of the meta-analysis and the low certainty of evidence, we have included an additional sentence in the Conclusions: "However, these findings should be interpreted with consideration of the limitations associated with pooled observational real-world data." (lines 529–530)
- Full table of results for each analysis (Supplementary table 13)

Certainty of evidence as assessed by GRADE was very low for all; however, this is to be expected given the observational nature of the included studies.